# Investigation of the Geopolymerization Potential of a Waste Silica-Rich Diabase Mud

**DOI:** 10.3390/ma15093189

**Published:** 2022-04-28

**Authors:** Maria Spanou, Sokrates Ioannou, Konstantina Oikonomopoulou, Pericles Savva, Konstantinos Sakkas, Michael F. Petrou, Demetris Nicolaides

**Affiliations:** 1Frederick Research Center, P.O. Box 24729, Nicosia 1303, Cyprus; res.mas@frederick.ac.cy (M.S.); d.nicolaides@frederick.ac.cy (D.N.); 2Department of Civil Engineering, Abu Dhabi Men’s Campus Higher Colleges of Technology, Abu Dhabi 25026, United Arab Emirates; sioannou@hct.ac.ae; 3Department of Civil and Environmental Engineering, University of Cyprus, 75 Kallipoleos Av., P.O. Box 20537, Nicosia 1678, Cyprus; oikonomopoulou.konstantina@ucy.ac.cy (K.O.); petrou@ucy.ac.cy (M.F.P.); 4Latomia Pharmakas, 23 Themistokli Dervi Av., S.TA.D.Y.L. Building, P.O. Box 23504, Nicosia 1066, Cyprus; 5RECS Civil Engineers & Partners LLC, 23 Themistokli Dervi Av., S.TA.D.Y.L. Building, P.O. Box 23504, Nicosia 1066, Cyprus; ksakkas@recsengineering.com

**Keywords:** waste diabase mud, geopolymer binders, alkali activated materials, materials valorisation, sustainability, mix design, early age compressive strength

## Abstract

Diabase mud (DM) is a silica-rich residue yielding from aggregate crushing and washing operations in quarries. This work focuses on identifying the geopolymerization potential of a diabase mud through characterization of its mineralogical composition, investigation of its reactivity, and assessment of the early compressive strengths of alkali activated mixtures formulated based on the mud’s dissolution results. The findings suggest that considerably low amounts of Al and Si metals were dissolved following the dissolution tests conducted on DM, however, the incorporation of small quantities of CEM I, gypsum, and metakaolin (MK) moderately at a Na_2_SiO_3_:NaOH ratio of 50:50 and with a molarity of NaOH of 4 M enhanced the geopolymerization compared to low L/S ratio mixtures cured at different conditions. When M was increasing, the high L/S ratio mixtures exhibited fluctuations in strengths, especially beyond a 10 M NaOH molarity. Maximum strengths of mixtures at equivalent molarity of 10 were achieved when the Na_2_SiO_3_:NaOH ratio reached 30:70, regardless of the ambient conditions and the presence of CEM I. The curing conditions, the ratio of Na_2_SO_3_:NaOH, and the presence of CEM I in the DM-based mixtures did not appear to significantly affect the mixture when NaOH concentration was between 2 M and 4 M; at higher molarities, however, these enhanced the strengths of the geopolymerized DM.

## 1. Introduction

The exponential demand for cement production–which currently reaches over 4.3 billion tonnes per annum worldwide [1]–imposes considerable pressure on the industries towards developing initiatives for minimizing the environmental impact associated with the cement’s embodied CO_2_ emissions. Backed by the desire to deliver sustainable and cost-effective solutions in construction, the conversion of waste into secondary raw materials is an imperative initiative within the framework of the Circular Economy agenda for the European Union [2]. In this perspective, one of the most promising methods for utilizing byproducts and industrial waste is the geopolymerization. Within this technology, the reaction mechanism to form inorganic polymers is essentially based on an alkali-based activation i.e., dissolution of the solid precursor by alkaline hydrolysis, gelation, and finally rearrangement and reorganisation to a three-dimensional network that resembles a zeolitic-like structure. The alkaline activators are usually concentrated solutions of Na,K–hydroxides and Na,K–silicates, whereas most typically used solid precursors are Al-Si- and Ca-Si-rich materials in nature. Prominent examples of such precursors are fly ash from coal combustion, metakaolin, and ground granulated blast furnace slag. Since these byproducts are associated with a considerably low embodied carbon footprint (for example, 4 kg of CO_2_ emitted per tonne of fly ash produced; 52 kg of CO_2_ emitted per tonne of slag produced [3]), the alkali activation of these materials certainly offers environmental advantages as well as promising durability, both aspects that have been indeed extensively researched [4,5,6,7,8,9,10,11,12,13,14,15,16,17]. The particular technology, in what is known as alkali-activated cement technology, has unveiled a promising potential, especially when considering slag, fly ash, and metakaolin for developing low carbon binders in non-structural applications, as featured in extensive research reviews [18,19,20,21,22].

In Cyprus, while slag and fly ash are in little to non-existent amounts due to the non-presence of steel industries or coal on the island. There is, however, an equally challenging issue to be addressed, and this is the large amounts of deposits of diabase mud (DM) accumulated as a sludge waste from the quarries. DM is a form of sludge waste that derives from solid fractions following aggregate crushing and/or aggregate washing operations. It is a residue yielded with a humidity content ranging between 25–30% and with a mineralogical composition consisting mainly of Al and Si oxides. This essentially makes it a promising candidate for potential alkali activation. At the same time, the accumulation rate of DM in quarries in Cyprus is indeed high, as certain plants were reported to exhibit an annual production of nearly 25,000 tonnes. Further, disposing of the mud in landfills within the production facilities is considerably limiting their storage capabilities. Considering the abovementioned issues, if a possible activation of the DM occurs to a sufficient degree so as to yield a durable, low carbon hydraulic binder, then there would be a leeway for the current issues of mud storage/accumulation/disposal in Cyprus to be controlled. At the same time, the possibility of sustainable solutions will be developed and offered to the market. Moreover, it would cater to the EU legislation related to recycling requirements of waste materials in Cyprus (which were introduced in 2012 [23]), thus ultimately enabling routes for the state to set regulations and targets related to industrial waste management.

The degree of the reactivity of the DM that is yielded from quarry operations for potential use in geopolymerization has not been researched to a considerable extent. Previous work [24,25,26] suggests that the quarry sludges or aggregate wash muds–depending on the oxide contents and reactivity–may scantily compete as a suitable precursor for potential alkali activation for yielding durable alkali-activated binders. There is, in addition, the important issue of the mineralogical variability of these residue muds due to the variations in the aggregate compositions existing in quarries in different geographic regions. Such variability, combined with additional key parameters that affect both the reactivity of the aluminosilicates (i.e., amorphism, fineness, specific surface area) as well as the precursor consumption (quantity and modulus of the alkali solutions for defining Si/Al or (Si + Al)/Na ratios of the mixtures), needs to be carefully taken into consideration when examining a by-product as a candidate for geopolymerization [5,6,7,8,9,10,11,12,13,14,15,16,17,18]. It is therefore critical to characterize and quantify the contents of key soluble elements, especially Al and Si, along with the crystallinity profile of the material and the corresponding phase amounts at the highest possible accuracy, so as to enable drawing conclusions on whether the material is capable of being geopolymerized—and thus defining its potential for use in developing sustainable and durable alkali-activated binders.

In light of the above, this work seeks to define the geopolymerization potential of a diabase mud received from a Cyprus quarry through a comprehensive experimental program, which consisted of characterizing, initially, the DM, followed by quantification of its reactivity potential through dissolution tests, and finally investigating of early mechanical properties of formulations developed based on the reactivity of the material.

## 2. Materials and Methods

### 2.1. Experimental Program

The experimental program consisted of three distinct stages, as described below:-Characterization of the DM material as received from quarry;-Examining the DM material’s reactivity through dissolution tests (i.e., determining the soluble Al, Si contents) as a basis for then establishing properly proportioned inorganic polymer formulations; and-Development of the above selected inorganic polymer formulations incorporating a combination of solids and alkaline solutions and investigating their stability through early age compressive strength tests at varying curing conditions and varying mixing parameters.

### 2.2. Characterization of the DM

The first stage of the experimental program, as received DM from Pharmakas Quarries in Cyprus, was initially characterized through X-Ray Diffraction Analysis (XRD), Thermogravimetric Analysis (TGA), Energy Dispersive X-Ray Fluorescence (ED-XRF), and density tests. For these tests, the DM was oven-dried at 110 °C until constant mass, followed by being crushed into fine particles using a Los Angeles abrasion machine at 2000 cycles and then sieved through a 0.063 mm sieve.

X-ray diffraction analysis was conducted using a BRUKER D8 X-Ray Diffractometer (BRUKER, Billerica, MA, USA) with Cu Kα1 (Ni filtered) radiation in the 2-theta range of 2° to 60° and on a 0.02°/s step. X-ray diffractogram of the material is shown in Figure 1. The composition, as observed by the peaks, appeared to be mainly crystalline, comprising of quartz (SiO_2_), albite (NaAlSi_3_O_8_ or Na_1.0–0.9_Ca_0.0–0.1_Al_1.0–1.1_Si_3.0–2.9_O_8_), labradorite (Ca, Na)(Al, Si)_4_O_8_, as well as minor traces of calcite (CaCO_3_).

The differential thermal/thermogravimetric (DT/TGA) analysis (Figure 2), which was conducted from less than 100 °C to 1100 °C at nitrogen atmosphere and at a rate of 15 °C/min, confirmed the disintegration of the abovementioned phases with mass losses occurring mainly within the dehydroxylation range (i.e., 1.75% mass loss of labradoride/portlandite at 500–650 °C), as well as within the decarbonation range (i.e., approximately 1.2% mass loss of labradorite/calcite at 700–800 °C). It has to be noted that the recorded initial mass loss was attributed to the loss of humidity, whereas the transformation of quartz was not observed during the analysis, as the phenomenon is taking place in higher temperatures.

For obtaining the oxide contents of the DM, ED-XRF analysis was conducted using a SPECTRO XEPOS elemental spectrometer, the results of which are shown in Table 1. The results were validated through Atomic Absorption Analysis (Fusion method), as well. By observing the oxide contents, the DM appeared to be a silicate-rich material with almost 41% SiO_2_ content, which, in combination with 11% of Al_2_O_3_, indirectly provided a good degree of reactivity potential and indicated a promising precursor for geopolymerization, even though its amorphousness profile may have seemed less favorable.

### 2.3. Density of DM

Density measurements were obtained by oven-drying the as-received sample at 105 °C until constant mass, followed by grinding the material using pestle and mortar in sizes less than 63 microns and using a Quantachrome pycnometer to determine the density of the powdered sample. Four measurements were obtained, and the average density was reported as 2362 kg/m^3^ (standard deviation = 25.2 kg/m^3^). This value was slightly lower than the average reported density of the aggregates crushed and processed in the particular local quarry in Cyprus.

### 2.4. Reactivity (Dissolution Tests) and Basis of Establishing Formulations

Dissolution tests were conducted on the basis of assessing the reactivity of DM. Reactivity is defined as the concentration of leached-out soluble elements (Si, Al) following a specific time frame. The test, therefore, provided the basis for establishing and developing inorganic polymer formulations incorporating solids with alkaline solutions for the subsequent experimental phase. The leaching reagents used were either NaOH or KOH at molarities of 2 M, 4 M, 6 M, 8 M, 10 M, 12 M, and 16 M, and at a liquid to solid ratio (L/S ratio) of 250. The test involved adding 0.2 g of the DM sample in 50 g leaching reagent in a tube which was shaken for 24 h, and then centrifuging the mixture for 15 min at 3000 rpm, followed by filtrating the supernatant at 0.45 micron membrane-filter and analyzing the residue through inductively coupled plasma atomic emission spectroscopy (ICP-AES). Results of the dissolution tests are shown in Figure 3. Based on the leached contents of both Al and Si metals, the highest dissolution percentage was noted at 12 M of NaOH or KOH molarity, beyond which, however, the efficiency of both reagents appeared to decline. The leaching amounts of both metals were higher when NaOH was used as the reagent compared to the case when KOH was used, although the contents were still very low to render the DM as a proper precursor for geopolymerization in both cases. The results from Figure 3 essentially suggest that DM may not be sufficiently dissolved in any of the two alkaline activators (i.e., a max. dissolution of 2.54%, which is considerably lower than typical Al-Si dissolutions of fly ash or metakaolin), and this is mainly attributed to the presence of high amounts of crystalline phases in DM that are unable to be dissolved, even in highly alkaline concentrations. A minimum, commonly suggested, dissolution percentage required to create a stable geopolymeric material based on previous research [18,19,27,28] is approximately 10%, which typically yields formulations of appreciable compressive strengths without the addition of other materials. A linear correlation, moreover, appears to exist [8,10,18,19] between the metal dissolution contents and the mechanical properties of the inorganic polymers; i.e., higher amounts of metal dissolutions were reported to lead to higher compressive strengths. Therefore, to establish appropriate inorganic polymer formulations that promote the dissolution of the metals, it was decided to incorporate small quantities of suitable hydraulic materials that promote greater amorphicity of phases within the matrix and thus enhance the geopolymerization of the DM. The materials selected are discussed in the following section.

### 2.5. Solids, Alkaline Solutions and Basis of Establishing Formulations

For achieving stability of the geopolymerized formulations, commercially available materials of portland cement (CEM I to EN 197-1:2020), gypsum (calcium sulfate dihydrate, CaSO_4_·2H_2_O to EN 13279-1:2008), as well as commercially available metakaolin (MK), were all strategically incorporated in the mixtures at defined percentages (thereon all referred to as ‘solids’). The solids were added on the basis of promoting amorphicity in the matrix and thus enhancing the geopolymerization of the DM.

For preparing the alkaline activator solution, a combination of NaOH and Na_2_SiO_3_ solutions were used at progressively varying Na_2_SiO_3_:NaOH ratios ranging from 10:90 to 90:10, although predominantly considered ones were the 50:50 and 40:60 ratios. The NaOH solution was prepared in molarities of 2 M, 4 M, 6 M, 8 M, 10 M, and 12 M by mixing solid pellets with appropriate amounts of deionized water (to maintain constant properties of water),–the amount of which was determined based on NaOH molar mass for producing the required concentrations–and then stirring the mixture for 5 min using a magnetic stirrer. The solution was insulated for 24 h in ambient conditions and mixed with a commercially available Na_2_SiO_3_ solution in the abovementioned percentages to develop the alkaline activator (thereon referred to as ‘liquid’). Properties of all solids and alkaline solutions used are shown in Table 2.

Three different cases of inorganic polymer mixtures were investigated:-Case 1 included the development of 12 mixtures of varying molarities (2–12 M, at 2 M increment) and varying NaOH/Na_2_SiO_3_ ratios for a liquid to solid (L/S) ratio of 0.69, incorporating DM, CEM I and gypsum and oven-cured for 48 h at 70 °C;-In Case 2, a total of nine mixtures were developed using DM, CEM I and MK at a low L/S ratio of 0.38, again with varying molarities (2–0 M, at 2 M increment) and varying NaOH/Na_2_SiO_3_ ratios, air-cured at 20 °C and 65% RH for 72 h prior to testing;-And lastly, a Case 3 was considered where a total of eight mixtures were developed using DM and MK at a low L/S ratio (0.38), again with varying molarities (2–10 M, at 2 M increment) and varying NaOH/Na_2_SiO_3_ ratios and being conditioned in the oven at 50 °C for 72 h prior to testing.

The nature of the DM (i.e., being a waste material with mineralogical variability due to the differences in the compositions of aggregates existing in quarries in different geographic regions of the island), in combination with the very limited previous references on the characterization and valorization of the material, demanded the investigation of a wide spectrum of factors potentially influencing its activation. Therefore, the three Cases mentioned above were selected after a considerable number of small-scale preliminary trial mixtures and combinations, based on the evaluation of the results related to the characterization of the DM and either previous experiences of the research team, or some similar cases reported in the literature [28]. It is important to be noted that it was not the intention of the research team to perform a direct comparison of the results obtained from the three Cases under investigation. Instead, the selection of the three Cases and the variation of the parameters were all aimed to reveal the factors that would positively affect the activation of the DM. Compressive strength values at 24 and 48 h were obtained for the mixtures in Case 1, whereas 72-h strengths were determined for the mixtures developed in Cases 2 and 3. All the varied parameters of the experimental program are shown in Table 3 below.

The paper aims to present, discuss and evaluate the results obtained from each of the three Cases under study, and reveal the factors that could enhance the alkali activation of the three hybrid mixtures. It should be noted that due to the nature and origin of the DM, as well as due to the lack of any previous knowledge related to the potential of the material to be valorized through geopolymerisation, a variety of experimental investigations were necessary to draw conclusions on this aspect. In a subsequent paper, the most promising mixtures developed based on the information summarised in this paper will be presented, along with the obtained physical and mechanical properties, durability characteristics, and optimized curing regimes.

### 2.6. Proportioning, Mixing, Casting, Curing, and Properties Tested

The inorganic polymers were prepared by carefully mixing the liquids and solids described in the previous section at the selected proportions shown in Table 3 for a total mixing duration of 5 min. The paste was then cast in copolymer-based cubic moulds (50 × 50 × 50 mm^3^) in accordance to ASTM C109–20, and was vibrated using a vibrating table for 1 min. The early age uniaxial compressive strengths of the samples were then determined using a 2000 kN CONTROLS hydraulic compression machine at either 24 and 48 h (for Case 1), or 72 h (for Cases 2 and 3), and at a loading rate of 0.2 MPa/s in accordance to ASTM C109–20 [20].

## 3. Results and Discussion

Average compressive strength developments and NaOH concentrations of all investigated formulations are shown in Table 3, along with the relevant mix proportions of solids and liquids incorporated in the mixtures. Relationships between compressive strengths, molarity and Na_2_SiO_3_:NaOH ratios of 50:50 and 40:60 for mixtures in Case 1 are shown in Figure 4, whereas the same relationships related to Cases 2 and 3 are shown in Figure 5, Figure 6 and Figure 7. The compressive strength values were determined as the averages of three samples for each value, and the standard deviations are included in the figures.

The results obtained from Case 1 showed that the relationship between compressive strengths and molarities at both Na_2_SiO_3_:NaOH tested ratios (i.e., 50:50 and 40:60) and at both ages (i.e., 24 h and 48 h), was overall inversely proportional, i.e., showcasing a declining rate, although however, changing the molarities caused fluctuations on the compressive strengths, especially in the mixtures having a Na_2_SiO_3_:NaOH ratio of 50:50. In these particular mixtures, when the highest molarity NaOH was incorporated in the liquids (i.e., 12 M), a significant strength loss (66% reduction) was recorded compared to its highest strength value, which was observed at a molarity of 4 M. Another interesting point obtained was that the mixtures with Na_2_SiO_3_:NaOH ratios of 50:50 exhibited approximately 10–20% higher 48 h compressive strengths than those of the 40:60 ratio, however, this occurred only up to molarity of 8 M; beyond which, the mixtures of 50:50 ratio suffered a drastic 48 h strength loss of 66% when molarity was increased from 8 M to 10 M, and these eventually became weaker than the mixtures of 40:60 ratio, with nearly 10% lower 48 h strengths.

The results in Figure 4 suggest that formulations of relatively high L/S ratio (i.e., 0.69) developed with a Na_2_SiO_3_:NaOH ratio of 50:50 and with a molarity of NaOH solution of 4 M exhibited the highest strengths from all compared formulations. However, the particular mixtures appeared to be highly sensitive to variations in the molarity, leading to significant fluctuations in strengths when molarities were progressively increased, and eventually suffering drastic strength losses beyond 10 M NaOH concentrations. In mixtures of a ratio of 40:60, the fluctuations in strengths were less significant when molarities were progressively increasing, although strengths were still declining across the tested molarities.

In Case 1 and according to Figure 4, a general trend of compressive strength decrease with the increase of the alkalinity was observed with minor differentiations in the strengths (within a range of 10–15%). This trend is attributed to the fact that the DM cannot be dissolved in the alkaline solution, according to the results obtained from the dissolution tests. Therefore, the alkali presence and the increase of molarity did not affect the materials’ geopolymeric formation and the development of strength. In contrast, the increase of alkalinity seems to decrease the compressive strengths, since no geopolymeric reaction is taking place. On the other hand, the mix designs for Case 1 contain two pozzolanic materials (i.e., cement and gypsum) that favour the presence of water for developing high strengths. In mixtures of lower alkalinity, the cement and gypsum hydration reactions become dominant (due to a higher amount of unbound water, i.e., water not participating in the alkaline solution), thus resulting in higher strengths, as recorded.

Results of investigations carried on mixtures of Cases 2 and 3 are shown in Figure 5, Figure 6 and Figure 7. It should be noted that, while observations were made and provided within the text, no correlations could be made between the two Cases (i.e., 2 and 3) due to the variation of multiple parameters governing each case, such as the curing regime, the mixture design, the ambient conditions, and also possible different activation mechanisms. As has been aforementioned, it was not the intention of this research to perform a direct comparison of the results obtained from the three Cases under study. Instead, the selection of these Cases and the variation of the parameters were aimed to reveal the factors that would positively affect the activation of the DM, and therefore to maximise the effective utilization of the DM content in mixtures.

Cases 2 and 3 are summarized in Figure 5, Figure 6 and Figure 7 together due to the common presence of metakolin in the two different formulations, their identical age of testing (i.e., 72 h), and also some similar trends observed in the obtained experimental results. It was observed that the progressive increase in NaOH molarity caused an increase in the 72 h strengths of both sets of mixtures with an almost identical effect on strength values for 2 M and 4 M, and then, at higher molarities, reaching up to 9.43 MPa (Case 2) and 7.53 MPa (Case 3), respectively. When the molarity value increased from 6 M to 8 M, both sets of formulations experienced their highest increase rate in strengths (121% and 93% increase for Cases 2 and 3, respectively). Beyond 8 M and towards 10 M, Case 3 mixtures still exhibited an appreciable increase in strengths, which was even higher than that observed from 4 M to 6 M. In the same concentrations, however, the mixtures of Case 2 were not associated with any significant increase, indicating an optimum concentration between 8–10 M for their highest achievable strengths. The results in Figure 5 and Figure 7 suggest that the addition n of small quantities (4%) of CEM I in low L/S ratio DM-MK mixtures (i.e., Case 2), when conditioned in air ambient temperatures, led to a more significant increase in strengths at NaOH concentrations between 6 M and 8 M within the 50:50 Na_2_SiO_3_:NaOH ratio, although this effect was ceased at concentrations beyond 8 M and towards 10 M. This difference can be also observed in Figure 6 when comparing strength values between the 6 M–8 M range. At the same figure, when incorporating NaOH of molarity 2 M and 4 M, both air-cured and oven-cured mixtures yielded almost identical strengths regardless of their different Na_2_SiO_3_ ratios and regardless of the presence of CEM I in the formulation. Beyond a concentration of 4 M, and at least until the 8 M, there is a significant increase in strengths, which appears to be considerably sharper in the Case 2 mixtures. However, such an increase was less significant beyond 8 M in Case 2, in contrast to Case 3 mixtures. An additional observation for Case 3 made in Figure 5 (right hand part of the graph) was that a low L/S ratio DM-MK mixture with a 20:80 alkaline solution and its NaOH molarity at 10 M yields approximately the same strength as that of a mixture at 30:70 alkaline solution with a NaOH molarity of 6 M.

When both Cases were compared at equivalent molarity of 10 M (Figure 7), the maximum achievable strengths were found at 30:70 Na_2_SiO_3_:NaOH ratio, regardless of the presence of CEM I and regardless of the ambient conditions. It can be also seen, on the same figure, that the absence of CEM I in Case 3 appeared to have enhanced the strengths more significantly when the ratio was tending from 10:90 towards 30:70 when compared to Case 2 (i.e., when containing CEM I).

Fundamentally, the increase in compressive strengths in both Cases may be attributed to the increase of the alkalinity, which is mainly attributed to the presence of metakaolin in the mixtures, which predominantly reacts with the alkaline activator. Moreover, in Cases 2 and 3, the addition of the sodium silicate solution seems to have a positive effect on the evolution of the compressive strength. The continuous increase of Si content enhanced the content of dissolved elemental silicon, and the increase of Si(OH)_4_ monomer promotes the formation of more -Si-O-Si- bonds, thus forming a more stable bond structure.

Generally, the higher the Si content, the more stable it is, since the chemical bond strength of -Si-O-Si- is higher than the corresponding of -Si-O-Al- and -Al-O-Al-, and therefore higher energy is required to break the particular bond.

With Si/Al ratio increasing up to a certain extent (i.e., Case 3), the content of elemental silicon dissolved in the system is much higher than that of aluminum. Meanwhile, a part of Si(OH)_4_ will form a dimer after condensation reaction and then react with Al(OH)_4_ to form a stable long chain (-Si-O-Al-O-Si-O-) PSS polymer (Si/Al = 2) or more stable long chain (-Si-O-Al-O-Si-O-Si-O-) PSDS polymer (Si/Al = 3). This phenomenon is not presented in Case 2, since the presence of cement creates C-S-H phases in parallel that are not enhanced with the presence of additional Si, while it in contrast intercepts the geopolymerisation process, and thus the compressive strength decreases (Figure 7).

## 4. Conclusions

The potential for geopolymerization of a diabase mud received from a Cyprus quarry was investigated in this paper and the following conclusions were drawn, based on the results:Considerably low amounts of Al and Si metals were dissolved following the dissolution tests conducted on DM, which essentially suggests that the material alone did not possess sufficient reactivity potential so as to give chemically stable alkali activated binders.Based on the above, mixtures of high L/S ratio incorporating CEM I, gypsum and MK at a Na_2_SiO_3_:NaOH ratio of 50:50 and with a molarity of NaOH of 4 M enhanced the geopolymerization of the DM, but appeared to be highly sensitive when M was increasing, leading to significant fluctuations in strengths and eventually suffering drastic strength losses beyond a 10 M NaOH molarity.The curing conditions, the ratio of Na_2_SO_3_:NaOH, and the presence of CEM I in the DM-based mixtures did not appear to significantly affect the compressive strengths when the NaOH concentration was between 2 M and 4 M (i.e., Cases 2 and 3); at higher molarities, however, these enhanced the strengths of the geopolymerized DM.Maximum strengths of mixtures at equivalent molarity of 10 were achieved when the Na_2_SiO_3_:NaOH ratio reached 30:70, regardless of the ambient conditions and regardless of the presence of CEM I.A low L/S ratio DM-Metakaolin mixture with a 20:80 alkaline solution with its NaOH molarity at 10M yields almost identical 72 h strength to that of a mixture at 30:70 alkaline solution with a NaOH molarity of 6 M.

## Figures and Tables

**Figure 1 materials-15-03189-f001:**
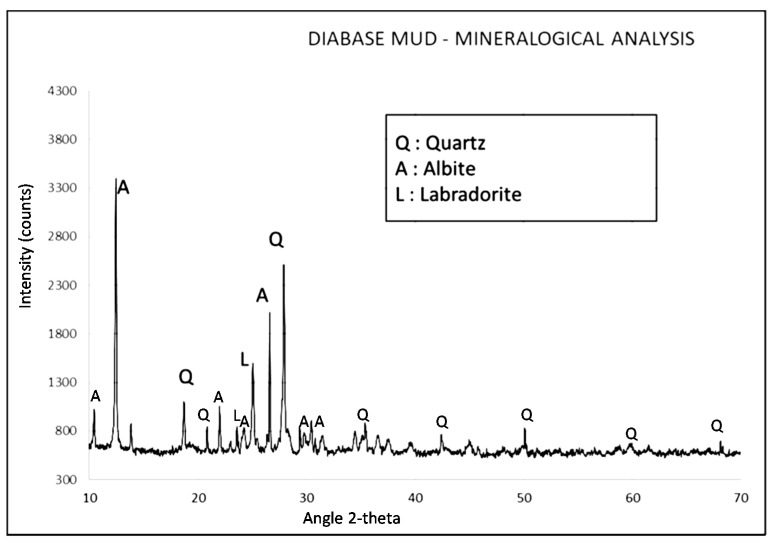
XRD diffractograms of the investigated DM material.

**Figure 2 materials-15-03189-f002:**
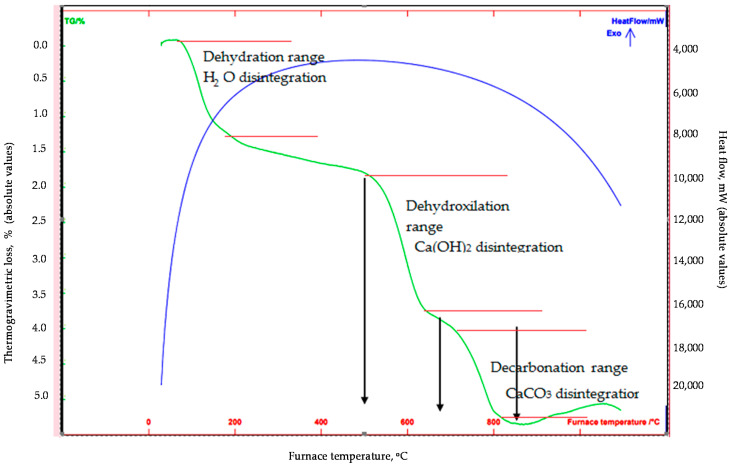
DT/TGA curves of the investigated DM material.

**Figure 3 materials-15-03189-f003:**
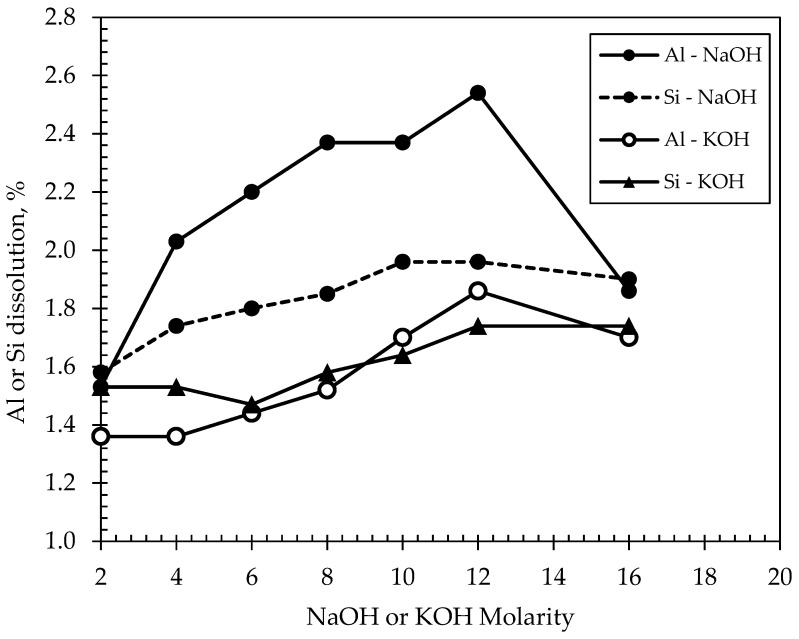
Results of dissolution of Al and Si elements of the investigated DM material.

**Figure 4 materials-15-03189-f004:**
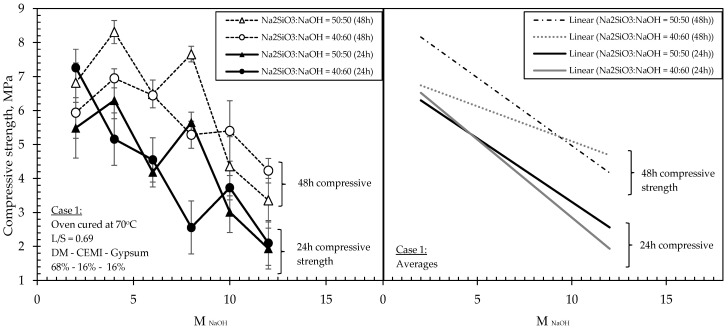
Effect of molarity of NaOH on the 24 h and 48 h compressive strength of formulations investigated in Case 1.

**Figure 5 materials-15-03189-f005:**
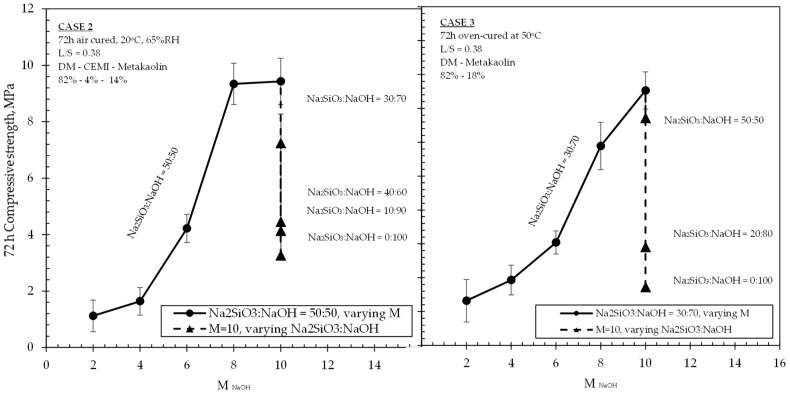
Effect of NAOH molarity and Na_2_SiO_3_: NaOH ratio on the 72 h strength of formulations at Cases 2 (**left**) and Case 3 (**right**).

**Figure 6 materials-15-03189-f006:**
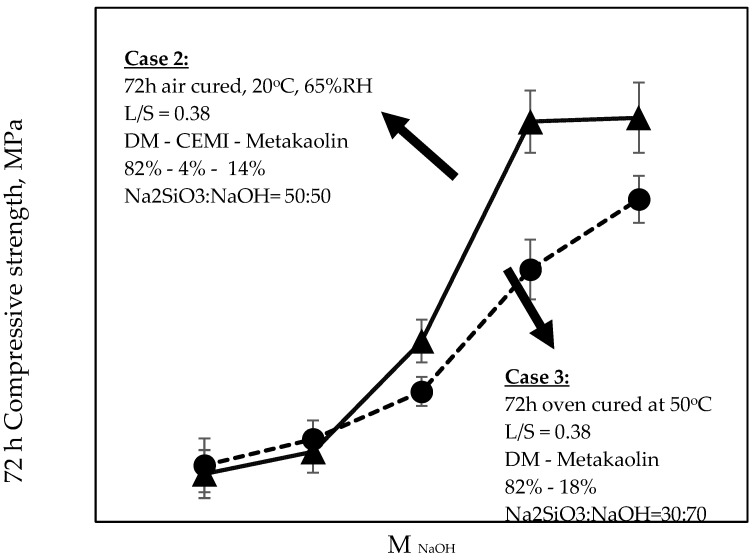
Relationship between 72 h compressive strength and NaOH molarity of formulations in Cases 2 and Case 3 at Na2SiO3:NaOH ratios of 50:50 and 30:70, respectively.

**Figure 7 materials-15-03189-f007:**
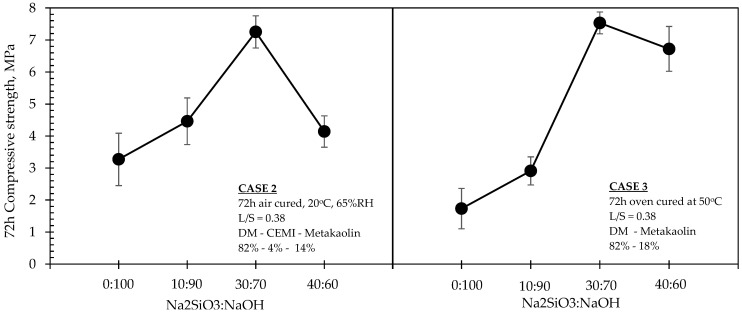
Effect of Na_2_SiO_3_: NaOH ratio on the 72 h strength of formulations in Cases 2 (**left**) and 3 (**right**), for M_NaOH_ = 10.

**Table 1 materials-15-03189-t001:** Oxide composition of the DM material from ED X-Ray Fluorescence (XRF) analysis.

	Na_2_O	MgO	Al_2_O_3_	SiO_2_	CaO	ZnO	FeO
**%**	2.53	8.87	11.14	40.91	5.36	1.71	13.65

**Table 2 materials-15-03189-t002:** Properties of alkaline solutions and solid used in the experimental.

Material	Conformity Standard	Nomenclature	Particle Size (Median, μm)	Specific Gravity	pH	Specific Surface (m^2^/kg)
Portland cement CEM I	EN 197-1:2020	CEM I	19.6	3.1	12.5	330
Metakaolin	NF P18-513	MK	1.2	2.5	5	14,200
Gypsum	EN 13279-1:2008	CaSO_4_·2H_2_O	24.5	2.82	7.2	260
Sodium Silicate solution	-	Na_2_SiO_3_	-	1.37	11.35	-
Sodium Hydroxide solution	-	NaOH	-	2.13	13	-

**Table 3 materials-15-03189-t003:** Mix proportions and properties of the developed formulations used in the experimental program.

Curing Condition	Formulation Nomenclature	L/S Ratio	DM % by wt. Solids	CEM I % by wt. Solids	Gypsum % by wt. Solids	MK % by wt. Solids	Na_2_SiO_3_:NaOH Ratio %:% by wt.	M_NaOH_	24 h Average Compressive Strength (MPa)	48 h Average Compressive Strength (MPa)	72 h Average Compressive Strength (MPa)
CASE1: 24 h or 48 h in the oven at 70 °C	M1	0.69	68	16	16	-	50:50	12	1.94	3.36	-
	M4						40:60	12	2.1	4.23	-
	M5						50:50	10	3.01	4.36	-
	M6						40:60	10	3.73	5.4	-
	M7						50:50	8	5.65	7.66	-
	M8						40:60	8	2.56	5.29	-
	M9						50:50	6	4.19	6.49	-
	M10						40:60	6	4.55	6.45	-
	M11						50:50	4	6.3	8.31	-
	M12						40:60	4	5.16	6.95	-
	M13						50:50	2	5.49	6.82	-
	M14						40:60	2	7.26	5.94	-
CASE 2: Air cured at 20 °C, 65% RH (72 h)	CyDIA-0007g	0.38	82	4	-	14	50:50	10	-	-	9.43
	CyDIA-0008g						40:60	10	-	-	4.14
	CyDIA-0009g						30:70	10	-	-	7.25
	CyDIA-0010g						10:90	10	-	-	4.46
	CyDIA-0011g						0:100	10	-	-	3.27
	CyDIA-0012g						50:50	8	-	-	9.34
	CyDIA-0013g						50:50	6	-	-	4.22
	CyDIA-0014g						50:50	4	-	-	1.64
	CyDIA-0015g						50:50	2	-	-	1.12
CASE 3: Oven cured 50 °C for 72 h	CyDIA-0004f	0.38	82	-	-	18	50:50	10	-	-	6.72
	CyDIA-0003f						30:70	10	-	-	7.53
	CyDIA-0002f						20:80	10	-	-	2.91
	CyDIA-0001f						0:100	10	-	-	1.73
	CyDIA-0004h						30:70	8	-	-	5.89
	CyDIA-0003h						30:70	6	-	-	3.04
	CyDIA-0002h						30:70	4	-	-	1.93
	CyDIA-0001h						30:70	2	-	-	1.32

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
