# Peer review of "Investigation of the Geopolymerization Potential of a Waste Silica-Rich Diabase Mud"

_materials, 2022, doi:10.3390/ma15093189_

Round 1

Reviewer 1 Report

Please find attached my comments

Author Response

All comments are addressed in the attached file.

Reviewer 2 Report

The paper "Investigation of the geopolymerization potential of a waste silica-rich diabase mud", although relevant and interesting, can only be considered for publication after major corrections, fundamental to the process:

(1) The abstract should be heavily revised, there is a lack of quantitative results and more detailed analyzes necessary for understanding, as well as its main conclusions;
(2) The manuscript in general has many formatting and editing errors, the figures must be proportionate in the text, in addition there are several citations errors!!
(3) The introduction should address other studies of more recent geopolymer materials, such as: 10.1016/j.cscm.2022.e00937; 10.1016/j.cscm.2021.e00839; 10.1016/j.jobe.2020.102010.
(4) Please note that a detailed description of materials, and determination of precursors and activators is required, as is the detail of this procedure;
(5) Authors should better explain the behavior of the resistance verified in terms of oscillations, a statistical analysis is necessary, as well as a better discussion of all the results;
(6) Conclusion is too extensive, authors should reduce its length.

Author Response

(The authors gave the same response as above.)

Round 2

Reviewer 2 Report

The authors made all the corrections suggested and the paper is now of a quality suitable for publication.